# Gut-derived peptidoglycan remotely inhibits bacteria dependent activation of SREBP by *Drosophila* adipocytes

**Bernard Charroux***, **Julien Royet** *

Aix-Marseille Université, CNRS, IBDM-UMR7288, Turing Center for Living Systems, Marseille, France

* bernard.charroux@univ-amu.fr (BC); julien.royet@univ-amu.fr (JR)

## Abstract

Bacteria that colonize eukaryotic gut have profound influences on the physiology of their host. In *Drosophila*, many of these effects are mediated by adipocytes that combine immune and metabolic functions. We show here that enteric infection with some bacteria species triggers the activation of the SREBP lipogenic protein in surrounding enterocytes but also in remote fat body cells and in ovaries, an effect that requires insulin signaling. We demonstrate that by activating the NF-κB pathway, the cell wall peptidoglycan produced by the same gut bacteria remotely, and cell-autonomously, represses SREBP activation in adipocytes. We finally show that by reducing the level of peptidoglycan, the gut born PGRP-LB amidase balances host immune and metabolic responses of the fat body to gut-associated bacteria. In the absence of such modulation, uncontrolled immune pathway activation prevents SREBP activation and lipid production by the fat body.

**Data Availability Statement:** All relevant data are within the manuscript and its Supporting information files.

**Funding:** This study was supported by the following grants: Investissements d'Avenir–Labex

## Author summary

An increasing body of evidence indicates that microbes, which live closely associated with animals, significantly influence their development, physiology and even their behavior. The mechanisms that underly these mutual interactions are not yet completely understood. Using *Drosophila* as a model system, we study the impact of gut bacteria on the host physiology. We present here data showing that some bacteria present in the fly gut can stimulate the production of lipids in the remote fat body tissue via gut autophagy and insulin signaling. However, these bacteria produce many compounds and metabolites such as the cell wall peptidoglycan. Our data show that by cell-autonomously activating the NF-κB signaling pathway in the remote fat body, cell wall peptidoglycan antagonizes bacteria-triggered lipogenesis. We finally show that to prevent this antagonistic effect, flies produce an enzyme, called PGRP-LB, that cleaves the peptidoglycan into its inactive form. Our data highlight the multiple layers of interactions that take place between gut-associated bacteria and a eukaryotic host.

INFORM, grant nr. ANR-11-LABX-0054, (https://anr.fr/) to JR; ANR PEPTIMET, grant nr. ANR-18-CE15-0018-02, (https://anr.fr/) to JR; Equipe Fondation pour la Recherche Médicale, grant nr. EQU201903007783, (https://www.frm.org/) to JR; Institut Universitaire de France (https://www.iufrance.fr/) to JR. The funders had no role in study design, data collection and analysis, decision to publish, or preparation of the manuscript.

**Competing interests:** The authors have declared that no competing interests exist.

## Introduction

In order to develop to adulthood and to later survive in their environment, multi-cellular organisms constantly adapt their metabolism needs to the nutrient availability. These nutrients come from food sources that are unavoidably contaminated by microbes on which they proliferate. Some of the microorganisms ingested with food, or those already associated with the digestive tract, directly participate to the host nutrition either by serving as food themselves or by metabolizing ingested aliments. These transient or permanent gut-associated microbes need to be either tolerated by the host if beneficial, or eliminated if detrimental, a function dedicated to the immune system. Hence, metabolism and immunity, that regulate the host's responses to these environmental inputs, nutriments and microbes, have co-evolved to provide a coordinated output at the organismal level. In mammals, this optimized response benefits from the fact that some immune cells are embedded into the adipose tissue [1–3]. Immune cells act as direct regulators of fat metabolism and innate immune signaling can impact metabolic responses cell-autonomously or via systemic inflammation [4–10]. Beside its role in lipid storage and energy expenditure, the adipose tissue is thus considered as an immune organ able to simultaneously sense nutrient and detect microorganism-derived compounds. Communication between the immune cells and adipocytes is essential to coordinate an *ad hoc* host metabolic response in physiological conditions and in response to microbial challenges [2].

In *Drosophila*, the fat body is the major site for lipid depository and combines energy storage, de novo synthesis, and breakdown functions that, in vertebrates, are dedicated to adipose and hepatic tissues [11,12]. In addition, via the production of many immune effectors including antimicrobial peptides, it plays a key role in orchestrating the innate immune responses to microbial infection [13–15]. Hence, *Drosophila* provides unique advantages to unravel the complex integration and regulation of these two essential physiological systems, before they evolved into more complex organs in vertebrates. Previous work has shown that *Drosophila* infection with bacteria or with the intracellular parasite *Tubulinosema ratisbonensis* leads to a depletion of fat body lipid stores [16]. Other studies, based on gain-of-function approaches, revealed that ectopic activation of the NF-κB pathways either Toll or IMD can result in lipid storage reduction. More precisely, immune signaling activation shifts anabolic lipid metabolism from triglyceride storage to phospholipid synthesis to support immune function [17].

Former results have shown that immune activation in the fat body cells can be triggered by bacteria present in the digestive tract. For that, the bacterial cell wall component peptidoglycan produced by gut-associated bacteria must cross the gut epithelium and reach the circulating hemolymph where it gets in contact with remote tissues. By activating receptors of the PGRP family expressed in adipocytes this gut-born bacterial ligand activates an NF-κB dependent AMP production [18–20]. This effect is buffered by the PGRP-LB amidase that, by cleaving the PGN into non-immunogenic fragments, prevents a diffusion of PGN to the hemolymph and hence a constant deleterious NF-κB activation in fat body cells of orally infected flies [20,21].

In the present study, we analyze the coordinate metabolic and immune responses of *Drosophila* to the presence of bacteria in the intestine. We show that flies orally fed with some bacteria species including *Escherichia coli* (*E. coli*) and *Erwinia carotovora carotovora* (*E.cc*) activate SREBP locally in enterocytes and remotely in adipocytes, in an insulin signaling-dependent manner. We also show that by activating the NF-κB/IMD pathway in adipocytes, PGN released by the same bacteria, cell-autonomously antagonizes SREBP-activation in adipocytes. Finally, we demonstrate that by regulating the levels of circulating PGN via the PGRP-LB amidase, flies can adjust their metabolic and immune responses towards gut bacteria.

## Results

### Specific gut bacteria species activate an SREBP-dependent lipogenesis in adult adipocytes

Our previous data showed that *Drosophila* enteric infection by the phytopathogen *E. cc* affects lipogenesis in adult adipocytes [20]. To further assess the effects of these bacteria of adult lipogenesis we monitored SREBP (Sterol Regulatory Element Binding Protein) activation in gnotobiotic flies orally infected with specific bacterial species (see Material and methods). SREBP is a conserved transcription factor that control lipid synthesis. Produced as a pro-peptide inserted into the endoplasmic reticulum (ER) membrane, it is post-translationally matured upon physiological modifications. In response to cellular lipids needs, SREBP exits the ER and travels to the Golgi apparatus where its active domain is freed by two successive proteolytic cleavages [22]. Upon nuclear translocation, mature SREBP activates the transcription of target genes controlling lipid synthesis [23]. We took advantages of the *Gal4*::*SREBP* reporter whose transcription relies on the native SREBP promoter and in which the resulting chimeric protein is processed like the endogenous SREBP [24]. We generated a novel *LexA*::*SREBP* reporter which mimics *Gal4*::*SREBP* activation (S1 and S2 Figs). Both chimeric fusion proteins are proteolytic cleaved and respectively activate UAS and LexAop fluorescent reporters, in cells wherein SREBP ensures *de novo* lipid synthesis [24–27] (Fig 1A and 1B, S1 and S2 Figs).

In addition to its constitutive activation in oenocytes already reported [24], *Gal4*::*SREBP* expression was unexpectedly detected in fat bodies of flies orally fed with a mixture of *E.cc* + sucrose compared to flies fed on sucrose only (Fig 1A and 1B). When *E. coli* was used to orally infect flies, an even stronger fat body SREBP activation was observed (Fig 1A and 1B). Other bacteria species such *Lactobacillus plantarum*[WJL] (*L. plantarum*[WJL]), *Acetobacter pomorum* (*A. pomorum*) or *Enterococcus faecalis* (*E. Faecalis*) failed to trigger activation of this lipogenic regulator (Fig 1A). These results were confirmed by immuno-histochemistry with an anti-SREBP antibody specific to the transcriptionally active N-terminal domain. While *E. cc* feeding induces a weak SREBP nuclear translocation in adipocyte, a much stronger response was observed in *E. coli*-fed adults fat bodies (Fig 1C). These results were further corroborated using the transcription of the SREBP target gene, *Acetyl-CoA synthase* (*ACS*) as a readout [24,28]. *ACS* mRNA levels were increased in *E.cc*-fed flies compared to sucrose-fed flies and this increase was even stronger with *E. coli* (Fig 1E).

### *E. coli* triggers SREBP activation in enterocytes and ovaries

We then asked whether *E.cc* and *E. coli* would activate lipogenesis in other tissues and organs known to be lipogenic. We first monitored the enterocytes which are in close proximity to the gut bacteria and represent another major source of lipids for the organism. Both *E. coli* and *E. cc*-triggered *Gal4*::*SREBP* activation in midgut enterocytes (Fig 1D). Gut bacteria also had some influences on female gonads. While ovaries of sucrose-fed females were atrophic, ovaries of females raised on an *E. coli*-contaminated solution resemble those of females fed on regular food, suggesting that *E. coli* represent a source of nutriment (Figs 2A and S3). Ovaries from *E.cc* females show an intermediate phenotype. In addition, *SREBP* activation was observed in midstage follicle of *E. coli* fed females, but not in ovaries from flies fed on sucrose only (Fig 2A and 2B). However, both gut and fat body of *E. coli* fed virgin females displayed a much weaker SREBP activation than mated ones (Fig 2C and 2D). This agrees with previous report showing that SREBP activation in enterocytes and in ovaries support oocyte production in mated females [29,30]. Consistently, males fed with *E. coli* displayed no sign of SREBP activation in enterocyte and only a constitutive no-bacteria dependent activation of SREBP in adipocytes (Fig 2C and 2D). All together our data show that gut-associated *E. coli* and *E.cc* activates SREBP

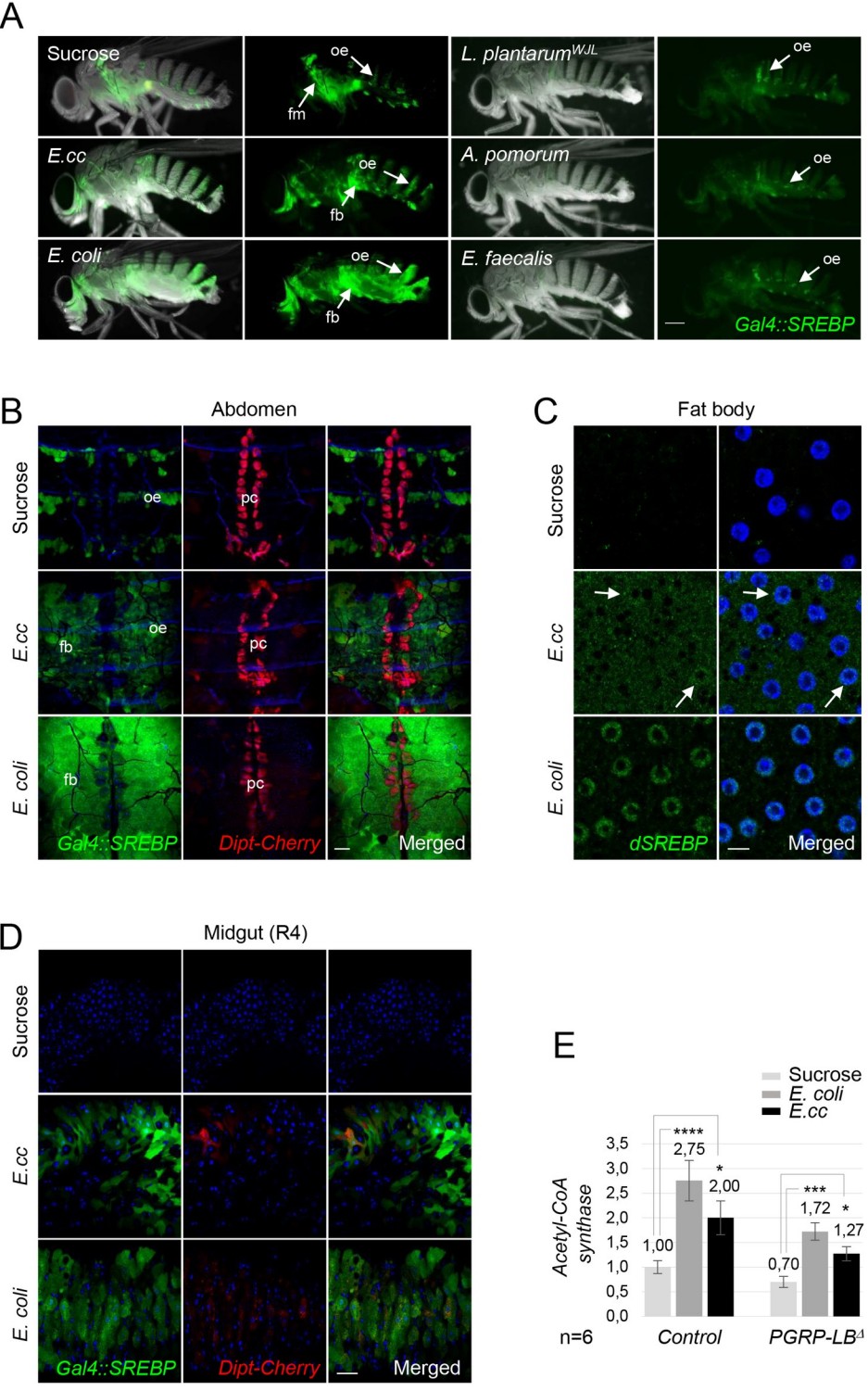

**Fig 1. Ingestion of *E. coli* or *E. cc* bacteria activates SREBP both in adipocytes and enterocytes.** (A) Pictures of adult flies fed 8 days with either sucrose or on a mixture of sucrose + bacteria such as *E. cc* or *E. coli* or *A. pomorum* or *E. faecalis* or *L. plantarum*[WJL], and showing *Gal4::SREBP* activation (green). Flies fed on sucrose show activation of *Gal4:: SREBP* in oenocytes and in flight muscle + oenocytes, respectively. *E. coli* feeding, and to a less extend *E. cc* ingestion, promote activation of *Gal4::SREBP* in fat bodies while ingestion of either *A. pomorum* or *E. faecalis* or *L. plantarum*[WJL] do not. Flies of the following genotypes were used: *w*[1118]/*w*[1118]; *Gal4::SREBP, UAS-2XEGFP/+*. oe: oenocytes, fm: flight muscles. (B) Confocal images of the dorsal part of adult abdominal carcasses viewed from inside, showing *Dipt-Cherry*

expression (red) and *Gal4::SREBP* activation (green) from flies fed on sucrose or on a mixture of sucrose + bacteria (*E. cc* or *E. coli*). Both *E. coli* and *E. cc* feeding activate *Gal4::SREBP* in fat body cells, while no expression of *Dipt-Cherry* is visible, as expected. Constitutive expression of *Dipt-Cherry* in pericardial cells and activation of *Gal4::SREBP* in oenocytes are indicated. Flies of the following genotypes were used: *w1118/w1118*; *Gal4::SREBP, UAS-2XEGFP/+; Dipt-CherryC1/Dipt-CherryC1*). pc: pericardial cells, oe: oenocytes. (C) Confocal images of fat body from adult flies fed 48h on sucrose, or on a mixture of sucrose + bacteria (*E. cc* or *E. coli*) showing immunofluorescence of *dSREBP* (green) and DAPI staining (blue). White arrows are showing nuclear translocation of *dSREBP* induced by *E. cc* feeding. Flies of the following genotypes were used: *w1118/w1118*. (D) Confocal images of the R4 domain of adult midguts from flies fed with either sucrose or with a mixture of sucrose + bacteria (*E. cc* or *E. coli*), showing *Dipt-Cherry* expression (red) and *Gal4::SREBP* activation (green). Both *E. coli* and *E. cc* feeding activates *Gal4::SREBP* in enterocytes, while a faint expression of *Dipt-Cherry* is induced. Flies of the following genotypes were used: *w1118/w1118*; *Gal4::SREBP, UAS-2XEGFP/+; Dipt-CherryC1/Dipt-CherryC1*). (E) Histograms showing the expression of *ACS* measured by q-RT-PCR and performed with mRNA extracted from adult abdominal carcasses of control or *PGRP-LBA* adults fed 4 days with either sucrose, on with a mixture of sucrose + bacteria (*E. coli* or *E. cc*). The mRNA level in non-infected control flies was set to 1 and values obtained with indicated genotypes were expressed as a fold of this value. Histograms correspond to the mean value ± SD of 6 experiments (n = 6). $^*p<0.05$, $^{***}p<0.001$, $^{****}p<0.0001$; Kruskal-Wallis test. Flies of the following genotypes were used: *w1118/w1118*;; *Dipt-CherryC1/Dipt-CherryC1* (*Control*) and *w1118/w1118*;; *PGRP-LBA, Dipt-CherryC1/ PGRP-LBA, Dipt-CherryC1* (*PGRP-LBA*). Scale bar is 0,25 mm (A), 100 μm (B), 5 μm (C) and 50 μm (D).

locally in enterocytes and remotely in fat body and ovaries suggesting that these bacteria represent a source of food for the flies.

## Bacterial ingestion promotes insulin signaling pathway

To further demonstrate that bacteria-mediated SREBP activation corresponds to a modification of the nutritional status the fly, we monitored insulin signaling in fat body cells using *tGPH* as cellular indicator of PI3K activity. Indeed, *tGPH* is recruited to plasma membranes by the second messenger product of PI3K, PI3P [31]. Flies fed with *E.cc* or *E. coli*, or raised on yeast extract as a medium containing AA source, showed *tGPH* membrane recruitment in adipocytes (Fig 3A). Such an effect was not observed when flies where fed on sucrose only (Fig 3A). These results, suggesting that gut *E. coli* and *E.cc* activate insulin signaling in adipocytes, were confirmed using q-RT-PCR on adult's abdominal carcasses. When compared to sucrose diet, mRNA levels of the negatively regulated insulin pathway target gene *4EBP/Thor* were decreased following *E. coli* and *E.cc* feeding, in a similar way as flies fed on regular food (Fig 3B). We then wondered whether insulin signaling was required for SREBP activation by gut bacteria using *chico1*, a loss-of-function allele of the Insulin Receptor Substrate Chico/IRS. We found that, in addition to their expected small size, *chico1* mutant females did not show any sign of *LexA::SREBP* activation when fed with *E. coli* (Fig 3C). The typical signal of *LexA:: SREBP* activation in enterocytes and in adipocytes of *E. coli* fed flies was absent in *chico1* mutants (Fig 3D and 3E). Only a weak, bacteria-independent, SREPB activation was observed in fat bodies and midguts from *chico1* animals (Fig 3D and 3E). Consistently, we found that mutants for Foxo, a catabolic transcription factor negatively regulated by the insulin pathway, displayed a consistent activation of SREBP in adipocytes when using a dose of *E. coli* (10 times less concentrated), that is normally not sufficient to activate *Gal4::SREBP* in wild-type flies (Fig 3F). Our results indicate that enteric infection by *E. coli* or *E. cc* activates insulin signaling, a prerequisite for SREPB activation in enterocytes and adipocytes. Consistently, overexpression of a dominant negative form of the insulin receptor specifically in the fat body was sufficient to completely block *E. coli* induced SREBP activation in adipocytes (Fig 3G).

## *E. coli*-dependent fat body SREBP activation depends on gut autophagy

Since gut bacteria-derived PGN can activate NF-κB/Relish in fat body cells and that Relish has been shown to restrain the transcription of the ATGL/Brummer lipase in the same cells [10],

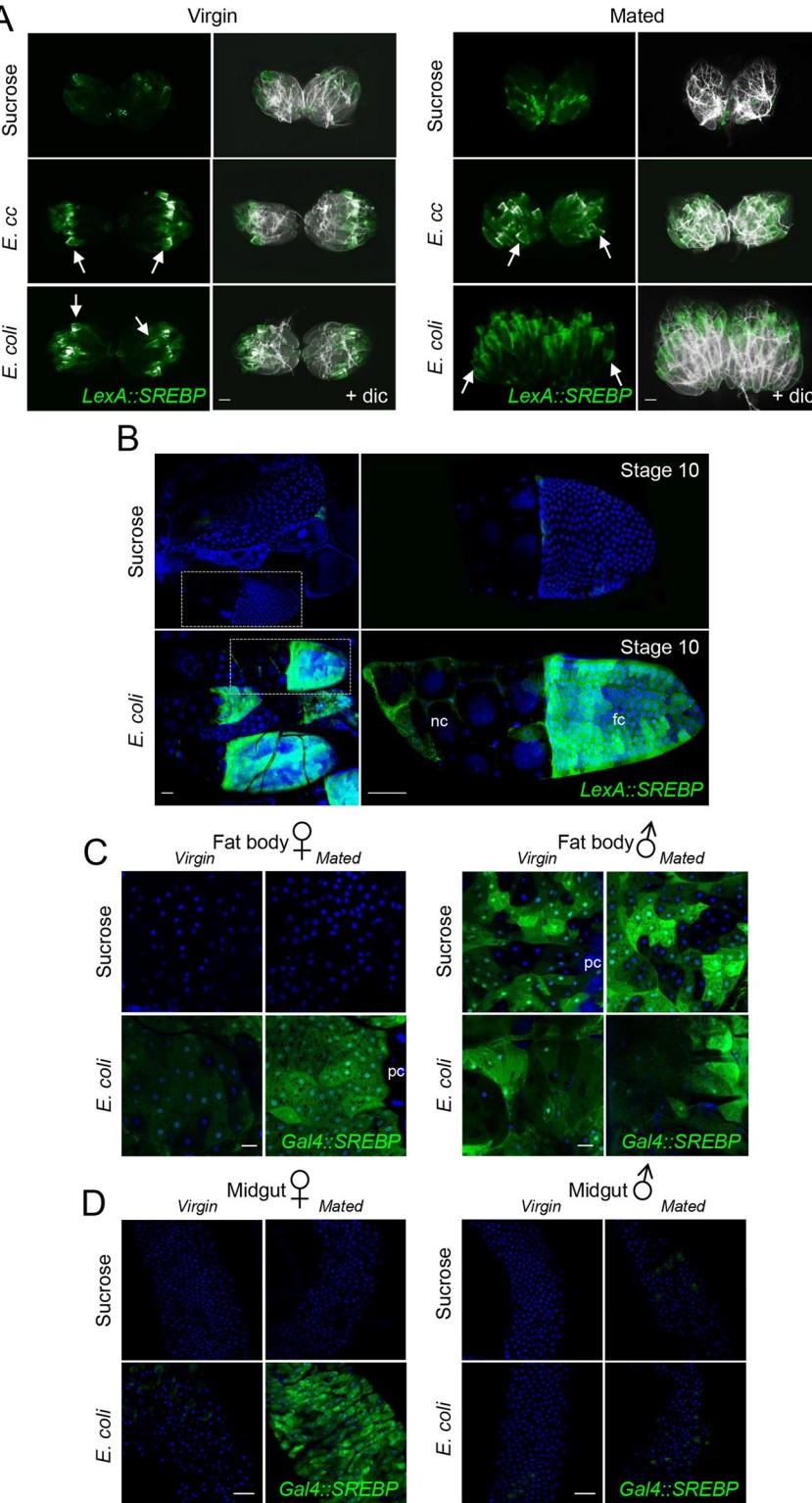

**Fig 2. *E. coli* ingestion triggers SREBP activation in ovaries and in fat body of mated females.** (A) Pictures of adult ovaries from virgin or mated females fed 2 days with sucrose or with a mixture of sucrose + bacteria (*E. cc* or *E. coli*), showing *LexA*::*SREBP* activation (green). Atrophy of ovaries is obvious in mated female fed with sucrose, not with *E. coli*, and to a less extend with *E. cc*. Stage 10 follicles display activation of *LexA*::*SREBP* (arrows in A), in females fed with *E.cc* or *E. coli*. (B) Confocal images of ovarioles dissected from females fed 2 days with sucrose or with *E. coli*,

showing *LexA::SREBP* activation (green). Females fed with *E. coli*. display activation of *LexA::SREBP* both in nurse cells and in follicle cells of stage 10 egg chambers. nc: nurse cells and fc: follicle cells. (C) Confocal images of fat body from virgin or mated females (or males) fed 2 days on sucrose or on a mixture of sucrose + *E. coli* showing *Gal4::SREBP* activation (green). Activation of *Gal4::SREBP* in adipocytes is strong in mated female and faint in virgin females. Male fat bodies however show no induction but constitutive activation of *Gal4::SREBP* in a *"salt and pepper"* pattern. (D) Confocal images of the R4 domain of adult midguts from virgin or mated females (or males) fed 2 days with a mixture of sucrose + *E. coli*, showing *Gal4::SREBP* activation (green). *E. coli* feeding does not promote activation of *Gal4::SREBP* in enterocytes from virgin females or males (virgin and mated). Flies of the following genotypes were used: *w^1118^/w^1118^, LexA::SREBP, 13XLexAop2-6XGFP/+* (A-B), *w^1118^/w^1118^; Gal4::SREBP, UAS-2XEGFP/+* (females in C-D), *w^1118^/Y; Gal4::SREBP, UAS-2XEGFP/+* (males in C-D). pc: pericardial cells, oe: oenocytes. Scale bar is 200 μm (A), 50 μm (B and D) and 20 μm (C and E).

we asked whether the bacteria-dependent SREBP activation could be consecutive to Brummer repression. In contrast to what was expected, *Brummer* transcript levels were increased in the presence of bacteria and followed a regulation that resembles that of *ACS* (Figs 3H and 1E). This suggests that SREBP activation in fat body of bacteria fed flies is not secondary to a reduced rate of lipid degradation by lipase. Another possibility is that SREBP activation is a direct consequence of an increase of the AA pool generated by bacteria-dependent gut protein catabolism. To test this hypothesis, we analyzed the consequences of blocking autophagy, known to participates to protein degradation, specifically in enterocytes on SREBP activation in adipocytes [32]. In contrast to what was observed in control flies, fat body from adults in which the autophagy effector protein ATG1 was down regulated via RNA interference in enterocytes, displayed a faint activation of SREBP upon *E. coli* feeding (Fig 3I).

## IMD signaling inhibits SREBP cell-autonomously in adipocytes

Although both *E. coli* and *E. cc* are able to activate fat body lipogenesis, we noticed that the effects were stronger with *E. coli* than with *E. cc*. Interestingly, previous works has shown that gut *E. cc* is a stronger inducer of fat body NF-κB signaling than *E. coli*, a difference attributed to the ability of *E. cc* to release PGN in larger amounts than *E. coli*. We hence hypothesized that bacteria-dependent gut-born PGN could buffer *SREBP* activation in fat body cells. To test this hypothesis, we analyzed the effects of enteric infection with bacteria in a mutant for PGRP-LB, an enzyme that cleaves PGN into non-immunogenic muropeptides. In such mutants, an excess of gut-born PGN reaches the different immune competent tissues leading to a higher NF-κB pathway activation. Fat body SREBP activation was weaker in *PGRP-LB* mutants than in wild type controls infected by *E. coli* (Figs 4A and 1B). This weaker SREBP activation was paralleled by a stronger NF-κB activation monitored with the *Dipt-mCherry* transgene (Fig 4A and 4C) or by q-RT-PCR (Fig 4B). This was, however, not the case in enterocytes (Fig 4D). These results were confirmed using isoform specific alleles of the PGN cleaving enzyme *PGRP-LB*. Inactivation of the extracellular isoform (PGRP-LB^PC^ named here PGRP-LB^sec^), which is expected to trigger an increase of circulating PGN levels, lead to an NF-κB signaling upregulation and a reduction of SREBP activation in fat body (Fig 4E). Such effects were not observed in flies carrying mutations in the cytosolic isoform (PGRP-LB^PD^ named here PGRP-LB^intra^) which do not affect the levels of circulating PGN. Moreover, the lack of *Gal4::SREBP* activation observed in *E. cc*-fed *PGRP-LB^Δ^* mutant flies, was reverted by the simultaneous inactivation of the IMD pathway core component *Dredd^F64^* (Fig 5A and 5B) demonstrating that an excessive IMD signaling can antagonize bacteria-dependent SREBP activation. The fact that similar results were obtained following PGRP-LC or Relish inactivation demonstrate that the steps implicating PGN sensing at the membrane and Relish-dependent transcription are involved in the process (S4 Fig).

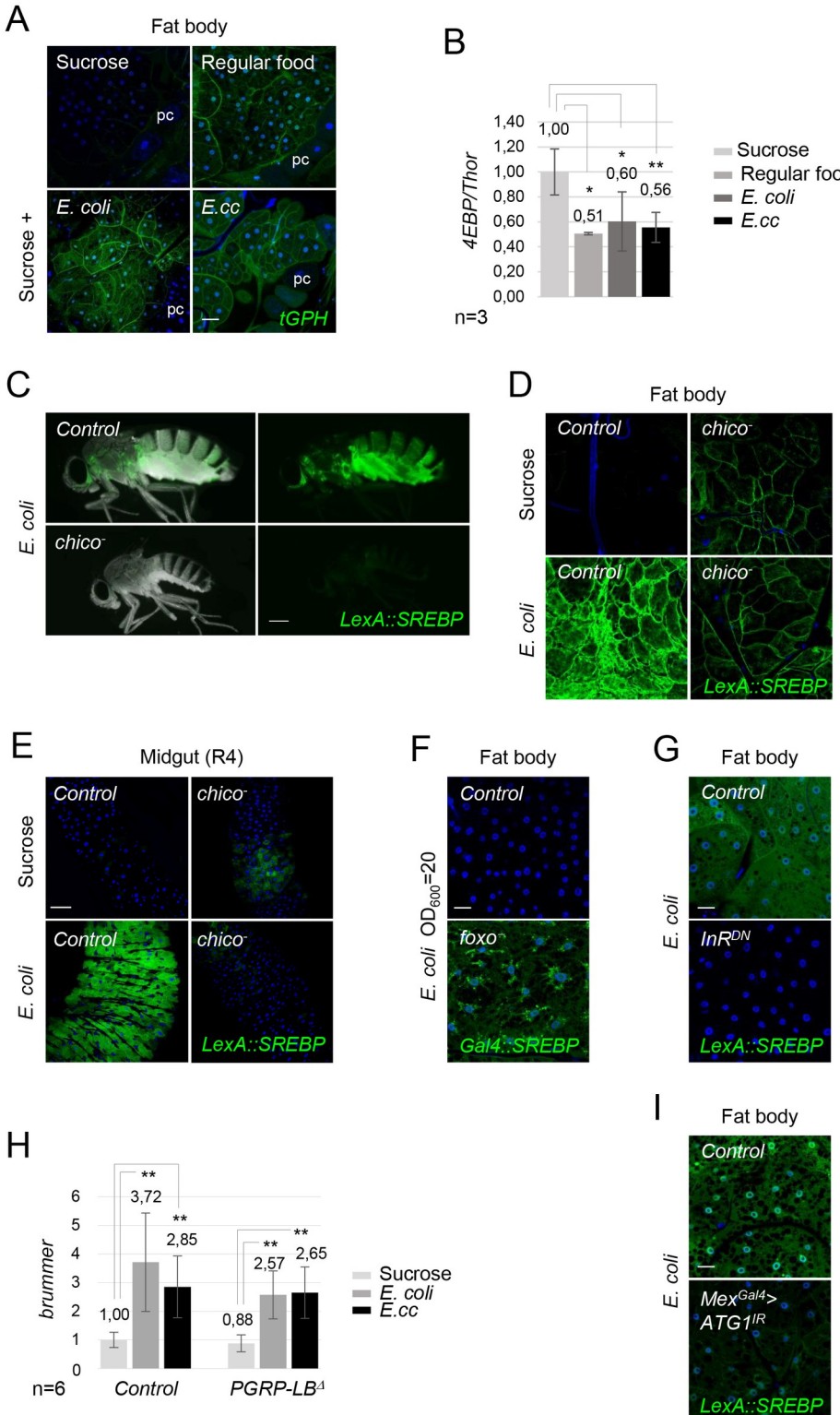

**Fig 3. _E. coli_ and _E. cc_ ingestion promotes systemic insulin signaling pathway and insulin signaling is required for SREBP activation by _E. coli_.** (A) Confocal images of fat body from flies fed 4 days with either sucrose, regular food or a mixture of sucrose + _E. coli_ or _E. cc_. The expression of the _tGPH_ marker is shown in green. An intense recruitment of _tGPH_ at the cell surface of adipocytes is observed in flies fed on regular food or orally infected with either _E. coli_ or _E. cc_, when compared to a sucrose diet. Flies of the following genotypes were used: $w^{1118}/w^{1118}$; _tGPH/tGPH_. pc:

pericardial cells. (B) Histograms showing the activation of the insulin signaling pathways measured by q-RT-PCR of *4EBP* mRNAs extracted from adult abdominal carcasses of control adults fed 4 days with either sucrose, regular food or a mixture of sucrose + bacteria (*E. coli* or *E. cc*). mRNA level in non-infected control flies was set to 1 and values obtained with indicated genotypes were expressed as a fold of this value. Histograms correspond to the mean value ± SD of three experiments (n = 3). *p<0.05, **p< 0.01; Kruskal-Wallis test. Flies of the following genotypes were used: $w^{1118}/w^{1118}$;; *Dipt-Cherry$^{C1}$/Dipt-Cherry$^{C1}$*). (C) Pictures of control or *chico$^-$* mutant flies fed 2 days with *E. coli* showing *LexA::SREBP* activation. Inhibition of insulin signaling pathway abolishes activation of *LexA::SREBP*. (D-E) Confocal images of fat body (D) or midgut R4 domain (E) from control or *chico$^-$* mutant flies fed 2 days with sucrose or with *E. coli*, showing *LexA::SREBP* activation. *E. coli* feeding does not promote activation of *LexA::SREBP* in adipocytes or enterocytes in absence of functional insulin signaling pathway. Flies of the following genotypes were used: $w^{1118}/w^{1118}$; *LexA::SREBP/Df(2L)ED729; 13XLexAop2-mcd8-GFP/+* (*Control* in C, D and E) and $w^{1118}/w^{1118}$; *LexA::SREBP, chico$^1$/Df(2L)ED729; 13XLexAop2-mcd8-GFP/+* (*chico$^-$* in C, D and E). (F) Confocal images of fat body from control or *foxo$^-$* mutant flies fed 2 days with a low dose (10x dilution) of *E. coli*, showing *Gal4::SREPB* activation. Foxo is a negative regulator of SREBP activation in adipocytes. Flies of the following genotypes were used: $w^{1118}/w^{1118}$; *Gal4::SREBP, UAS-2XEGFP/+; foxo$^{Δ94}$/+* (*Control* in F) and $w^{1118}/w^{1118}$; *Gal4::SREBP, UAS-2XEGFP/+; foxo$^{Δ94}$/foxo$^{25}$* (*foxo$^-$* in F). (G) Confocal images of fat body from control or mutant flies overexpressing a dominant negative version of InR (InR$^{DN}$) in adipocytes, and fed 2 days with *E. coli*. *LexA::SREBP* activation is shown in green. *E. coli* feeding does not promote activation of *LexA::SREBP* in absence of functional insulin signaling pathway in adipocytes. Flies of the following genotypes were used: $w^{1118}/w^{1118}$, *LexA::SREBP, 13XLexAop2-6XGFP/+; r4$^{Gal4}$/+* (*Control*) or $w^{1118}/w^{1118}$, *LexA::SREBP, 13XLexAop2-6XGFP/ UAS-InR$^{DN}$; r4$^{Gal4}$/Tub$^{Gal80ts}$* (*InR$^{DN}$*). (H) Histograms showing the expression of *brummer* measured by q-RT-PCR and performed with mRNA extracted from adult abdominal carcasses of control adults fed 4 days with either sucrose, on with a mixture of sucrose + bacteria (*E. coli* or *E. cc*). The mRNA level in non-infected control flies was set to 1 and values obtained with indicated genotypes were expressed as a fold of this value. Histograms correspond to the mean value ± SD of six experiments (n = 6). **p< 0.01; Kruskal-Wallis test. Flies of the following genotypes were used: $w^{1118}/w^{1118}$;; *Dipt-Cherry$^{C1}$/Dipt-Cherry$^{C1}$* (*Control*) and $w^{1118}/w^{1118}$;; *PGRP-LB$^Δ$, Dipt-Cherry$^{C1}$/PGRP-LB$^Δ$, Dipt-Cherry$^{C1}$* (*PGRP-LB$^Δ$*). (I) Confocal images of fat body from control or mutant flies expressing *UAS-ATG1$^{IR}$* in enterocytes, and fed 2 days with *E. coli*. *LexA::SREBP* activation is shown in green. *E. coli* feeding promotes a faint activation of *LexA::SREBP* when autophagy is blocked in enterocytes. Flies of the following genotypes were used: $w^{1118}/w^{1118}$, *LexA::SREBP, 13XLexAop2-6XGFP/Mex$^{Gal4}$; Tub$^{Gal80t}$/+* (*Control*) or $w^{1118}/w^{1118}$, *LexA::SREBP, 13XLexAop2-6XGFP/UAS-ATG1$^{IR}$, Mex$^{Gal4}$; Tub$^{Gal80t}$/+* (*Mex$^{Gal4}$>ATG1$^{IR}$*). Scale bar is 20 μm (A), 0,25 mm (C), 20 μm (D, F, G and I) and 50 μm (E).

To identify the tissue(s) in which NF-κB activation is required for SREBP regulation, we monitored the effects of IMD pathway components over expression, which is sufficient to activate downstream signaling in the absence of bacteria. *PGRP-LCa* or *IMD* overexpression, either ubiquitously (*da$^{Gal4}$*) or specifically in the fat body (*r4$^{Gal4}$*), prevented the activation of *LexA::SREBP* in *E. coli* fed flies (Fig 5C). Moreover, clonal over expression of IMD or *PGRP-LCa* cell-autonomously prevented *LexA::SREBP* activation in fat body of females fed with *E. coli* (Fig 5D). These data demonstrate that the gut bacteria-dependent SREBP activation in fat body cells is cell-autonomously repressed by a PGN-dependent IMD/NF-κB pathway activation.

## Inhibiting IMD signaling in adipocytes improves survival of *E. cc* infected *PGRP-LB$^Δ$* mutant flies

To test the physiological relevance of this antagonism, we monitored the survival curves and lipid droplet accumulation in various genetic combinations chronically infected with *E. cc*. As expected, whereas wild-type flies fed with *E. cc* showed lipid droplet accumulation, this was not the case for *PGRP-LB* mutants (Fig 6A). In addition, *PGRP-LB* mutants died much earlier than their wild-type siblings upon *E.cc* chronic infection (Figs 6B and S6). As shown for SREBP activation (Fig 5A), the reduced lifespan and lipid droplet non-accumulation in adipocytes observed in *E.cc*-infected *PGRP-LB$^Δ$* mutants were restored by the simultaneous inactivation of IMD pathway component *Dredd* (Fig 6A and 6B). Since IMD/NF-κB signaling specifically inhibits SREBP activation and lipogenesis in adipocytes, we tested whether IMD signaling buffering in adipocytes could ameliorate the survival of *PGRP-LB$^Δ$* mutant flies chronically infected with *E.cc*. To do so, we took advantage of a *UAS-dFadd$^{IR}$* transgene whose

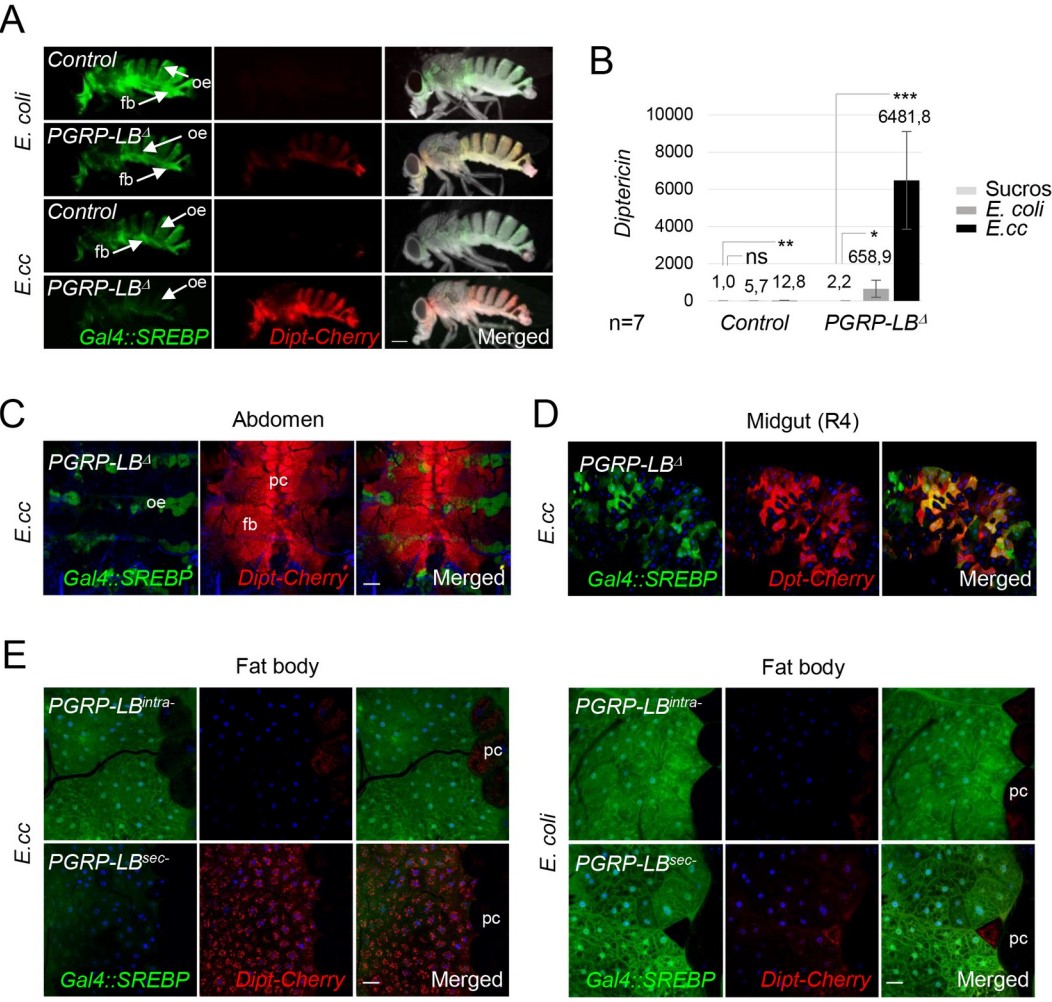

**Fig 4. Bacteria-dependent gut-born PGN antagonizes *SREBP* activation in adipocytes.** (A) Pictures of adult flies, control or *PGRP-LB^Δ* mutants, fed 8 days with a mixture of sucrose + *E. coli* or *E. cc*, and showing *Dipt-Cherry* expression (red) and *Gal4::SREBP* activation (green). *E. coli* feeding promotes activation of *Gal4::SREBP* in fat bodies from control and *PGRP-LB^Δ* mutant. Ingestion of *E. cc*, however, triggers activation of *Gal4::SREBP* in fat body from control flies, but not from *PGRP-LB^Δ* mutant's flies. As expected, activation of *Dipt-Cherry* is observed in fat body from *PGRP-LB^Δ* mutants fed with bacteria. The constitutive activation of *Gal4::SREBP* in oenocytes is indicated (oe arrows). Flies of the following genotypes were used: *w^1118/w^1118*; *Gal4:: SREBP, UAS-2XEGFP/+*; *Dipt-Cherry^C1/Dipt-Cherry^C1* (*Control*) and *w^1118/w^1118*; *Gal4::SREBP, UAS-2XEGFP/+*; *Dipt-Cherry^C1, PGRP-LB^Δ/Dipt-Cherry^C1, PGRP-LB^Δ* (*PGRP-LB^Δ*). (B) Histograms showing the expression of *Diptericin* measured by q-RT-PCR and performed with mRNA extracted from adult abdominal carcasses of control adults fed 4 days with either sucrose, on with a mixture of sucrose + bacteria (*E. coli* or *E. cc*). The mRNA level in non-infected control flies was set to 1 and values obtained with indicated genotypes were expressed as a fold of this value. Histograms correspond to the mean value ± SD of seven experiments (n = 7). *p<0.05, **p< 0.01, ***p<0.001; Kruskal-Wallis test. Flies of the following genotypes were used: *w^1118/w^1118*;; *Dipt-Cherry^C1/Dipt-Cherry^C1* (*Control*) and *w^1118/w^1118*;; *PGRP-LB^Δ, Dipt-Cherry^C1/PGRP-LB^Δ, Dipt-Cherry^C1* (*PGRP-LB^Δ*). (C-D) Confocal images of the dorsal part of adult abdominal carcasses viewed from inside (C) or of the midgut R4 domain (D) from *PGRP-LB^Δ* mutant flies fed with *E. cc*, showing *Dipt-Cherry* expression (red) and *Gal4::SREBP* activation (green). Adipocytes from *PGRP-LB^Δ* mutant flies display high level of *Dipt-Cherry* expression but no activation of *Gal4::SREBP*. Constitutive expression of *Dipt-Cherry* in pericardial cells and activation *Gal4::SREBP* in oenocytes are indicated (C). Enterocytes of *PGRP-LB^Δ* mutant flies display activation of both reporters (D). Flies of the following genotypes were used: *w^1118/w^1118*; *Gal4::SREBP, UAS-2XEGFP/+*; *Dipt-Cherry^C1, PGRP-LB^Δ/Dipt-Cherry^C1, PGRP-LB^Δ* (*PGRP-LB^Δ*). (E) Confocal images of adult fat body from CRISPR mutant flies *PGRP-LB^intra-* or *PGRP-LB^sec-*, fed 72h with *E. cc* (A) or *E. coli* (B) and showing *Dipt-Cherry* expression (red) and *Gal4::SREBP* activation (green). (A) *E. cc* feeding induces activation of *Dipt-Cherry* in adipocytes from *PGRP-LB^sec-* animals, but not from *PGRP-LB^intra-* ones. *Gal4::SREBP* activation is faint in the CRISPR-specific mutant allele *PGRP-LB^sec-* and strong in the *PGRP-LB^intra-* one. (B) *E. coli* feeding induced a comparable activation of *Gal4::SREBP* in both *PGRP-LB^sec-* or *PGRP-LB^intra-* adipocyte's mutant flies, but no activation of *Dipt-cherry*. Flies of the following genotypes were used: *w^1118/w^1118*;; *PGRP-LB^PD*Z10e, Dipt-Cherry^C1/PGRP-LB^PD*Z10e, Dipt-Cherry^C1* (*PGRP-LB^intra-*) and *w^1118/w^1118*;; *PGRP-LB^PC*10A, Dipt-Cherry^C1/PGRP-LB^PC*10A, Dipt-Cherry^C1* (*PGRP-LB^sec-*). pc: pericardial cells. Scale bar is 0,25 mm (A), 100 μm (C), 50 μm (D) and 20 μm (E).

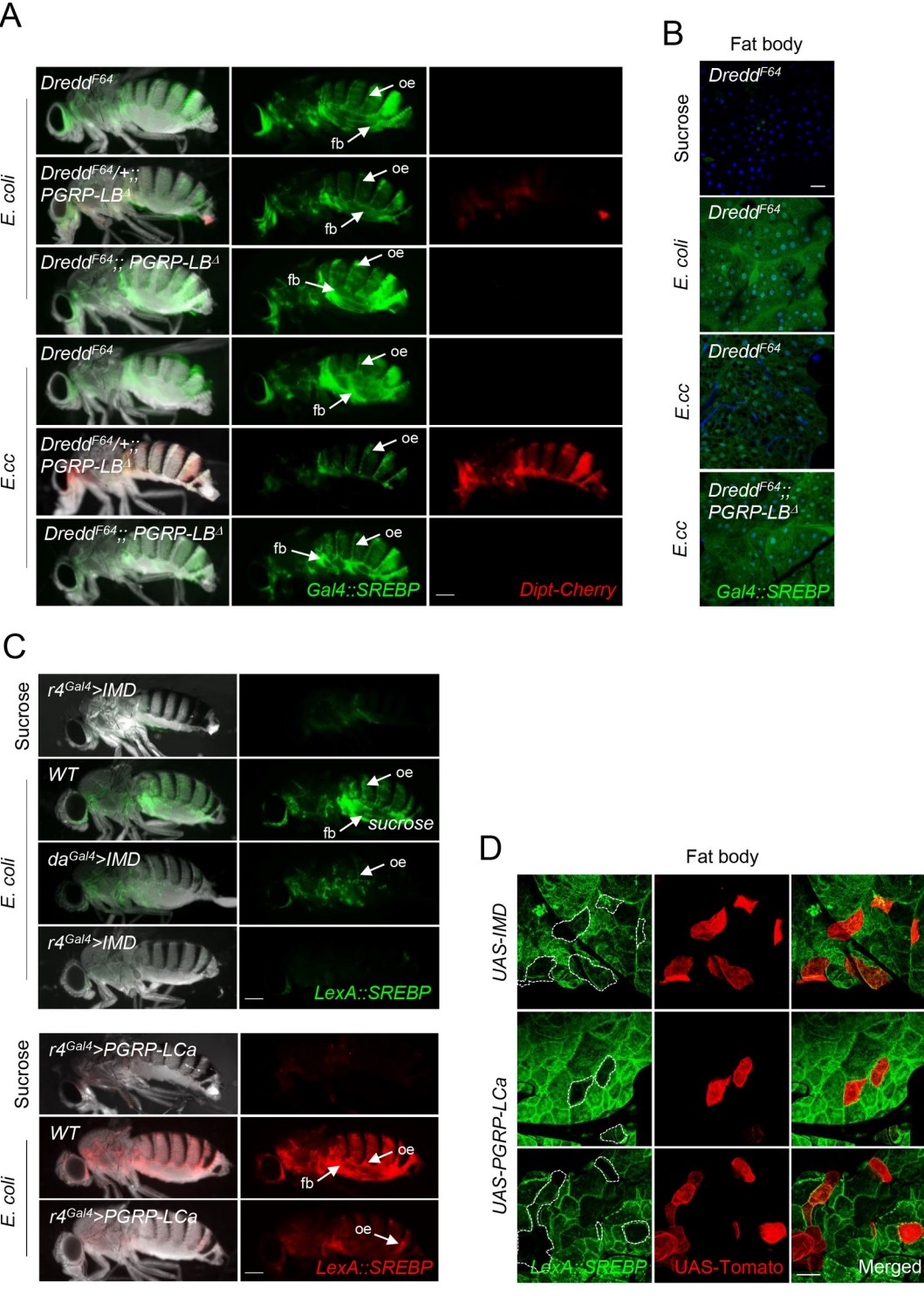

**Fig 5. The NF-κB signaling pathway inhibits SREBP activation cell-autonomously in adipocytes.** (A) Pictures of adult flies fed 7 days with either *E. coli* (A) or *E. cc* (B) showing *Dipt-Cherry* expression (red) and *Gal4*::*SREBP* activation (green). Flies mutant for the loss-of-function allele *Dredd^F64* activate *Gal4*::*SREBP* in fat body, when fed with either *E. coli* or *E. cc*. Two copies of the *Dredd^F64* allele suppress the faint and the lack of activation of *Gal4*::*SREBP* observed in *PGRP-LB^Δ* mutant flies fed with *E. coli* and *E. cc*, respectively (A). (B) Confocal images of fat body from flies fed 2 days with sucrose or with a mixture of sucrose + the indicated bacteria. Activation of *Gal4*::*SREBP* (shown in green) absent in sucrose fed flies, is similarly detectable in *Dredd^F64* mutant flies fed with either *E. coli* or *E. cc*. Double mutant *Dredd^F64*;; *PGRP-LB^Δ* flies activate *Gal4*::

*SREBP* in adipocytes when fed 2 days with *E. cc*. Flies of the following genotypes were used: *w^1118^, Dredd^F64^/ w^1118^, Dredd^F64^; Gal4::SREBP, UAS-2XEGFP/+; Dipt-Cherry^C1^/Dipt-Cherry^C1^* (for *Dredd^F64^*) or *w^1118^, Dredd^F64^/ w^1118^; Gal4::SREBP, UAS-2XEGFP/+; Dipt-Cherry^C1^, PGRP-LB^A^/Dipt-Cherry^C1^, PGRP-LB^A^* (for *Dredd^F64^/+;; PGRP-LB^A^*) or *w^1118^, Dredd^F64^/ w^1118^, Dredd^F64^; Gal4::SREBP, UAS-2XEGFP/+; Dipt-Cherry^C1^, PGRP-LB^A^/Dipt-Cherry^C1^, PGRP-LB^A^* (for *Dredd^F64^;; PGRP-LB^A^*). (C) Pictures of adults flies fed 3 days with sucrose or a mixture of sucrose + *E. coli* showing *LexA::SREBP* activation (green in top panels and red in bottom panels). Flies fed on sucrose and overexpressing IMD or PGRP-LCa in fat body cells (with *r4^Gal4^*), do not activate *LexA::SREBP*. Upon *E.coli* feeding, the activation of *LexA::SREBP* is inhibited in fat bodies from flies over expressing IMD or PGRP-LCa in fat body cells with *r4^Gal4^*, or from flies over expressing IMD ubiquitously with *da^Gal4^*. Flies of the following genotypes were used: *w^1118^/w^1118^, LexA::SREBP, 13XLexAop2-6XGFP/+; r4^Gal4^/+* (*Control* in top panels) or *w^1118^/w^1118^, LexA::SREBP, 13XLexAop2-6XGFP/+; da^Gal4^ or r4^Gal4^/UAS-IMD* (*da^Gal4^* and *r4^Gal4^* in top panels), or *w^1118^/w^1118^, LexA::SREBP, LexOp-mCherry.mito/+; r4^Gal4^/+* (*Control* in bottom panels) or *w^1118^/w^1118^, LexA::SREBP, LexOp-mCherry.mito/+; r4^Gal4^/UAS-PGRP-LCa* (*r4^Gal4^* in bottom panels). (B) Confocal images of fat body showing clones of adipocytes overexpressing IMD or PGRP-LCa (red) and activation of *LexA::SREBP* (green), from flies fed 2 days with *E. coli*. Fat body clones over expressing either IMD or PGRP-LCa inhibits *LexA::SREBP* activation is a strictly autonomous manner. Flies of the following genotypes were used: *w^1118^, CoinFLP^Gal4^/w^1118^; LexA::SREBP, LexOp-CD8-GFP-2A-CD8-GFP, UAS-CD4::Tomato/hs-FLP.G5, Tub^Gal80ts^; UAS-IMD or UAS-PGRP-LCa/+*. Scale bar is 0,25 mm (A and C) and 20 μm (B and D).

targeted expression can block IMD signaling over activation typically observed in guts and fat bodies of *E. cc*-infected *PGRP-LB^A^* mutant (Fig 6C). We then tested the effects of a tissue-specific IMD silencing on the lifespan of chronically *E. cc*-infected *PGRP-LB^A^* mutants. We found that blocking IMD signaling in enterocytes with *Mex^Gal4^* or in muscles using *Mef2^Gal4^* did not ameliorate the lifespan of *E. cc* fed *PGRP-LB^A^* mutants. In contrast, ubiquitous and fat body specific expression of *UAS-dFadd^IR^* significantly improved *PGRP-LB^A^* mutant resistance to chronic *E. cc* infection (Figs 6D and S6). These results suggest that by buffering IMD pathway activation in the fat body, the PGRP-LB amidase allows this tissue to generate lipids. This could be a mean for the host to better resist to chronic infection (Fig 7).

## Discussion

We showed here that gut-associated bacteria can influence host lipid metabolism by activating SREBP in adipocytes. Axenic flies fed with sucrose displayed phenotypes of undernourished animal, such as ovarian atrophy and reduced systemic insulin signaling. At contrary, *E. coli* fed animal had fully developed ovaries and displayed local (gut) and systemic (fat body) activation of the insulin signaling, genetically upstream of SREBP activation in these lipogenic organs. Which bacteria associated metabolites or constituents mediate these effects? Dietary amino-acids are obvious candidates [33,34]. Consistently, heat-killed *E. coli* remained good inducer of adipocytes SREBP (S5A Fig). In addition, gut specific inhibition of autophagy prevented its activation. These results, which strongly suggest that bacteria derived amino-acids are the triggering signal, are consistently with published data obtained using yeast as a food source [35]. Likewise, fly microbiota contributes to protein processing upstream of the nutrient-sensing Tor and insulin signaling to promote growth [36,37]. In addition, starved adult flies favored a bacteria-contaminated sucrose solution over an axenic one [38]. Together with other reports, our data show that flies can use microbes as amino-acids source leading to the activation of the insulin/Tor signaling and SREBP in fat body [35,39,40]. SREBP activation could be used by the host as a mean to replenish its lipid pools affected by the bacterial infection. It was recently shown that in infected flies, hemolymphatic lipids are removed by the Malpighian tubules and excreted [41]. This mechanism, which provided protection against lipid peroxidation, is a central component of host physiological adaptation to infection, since flies lacking it succumb to infection. Lack of SREBP activation in *Ecc* infected fat body would prevent lipid pools replacement and hence contribute to host death.

*L. plantarum^WJL^, A. pomorum, E. faecalis* and *Saccharomyces cerevisiae* were not able to activate SREBP, in agreement with previous results showing that *S. cerevisiae* and *L. plantarum*

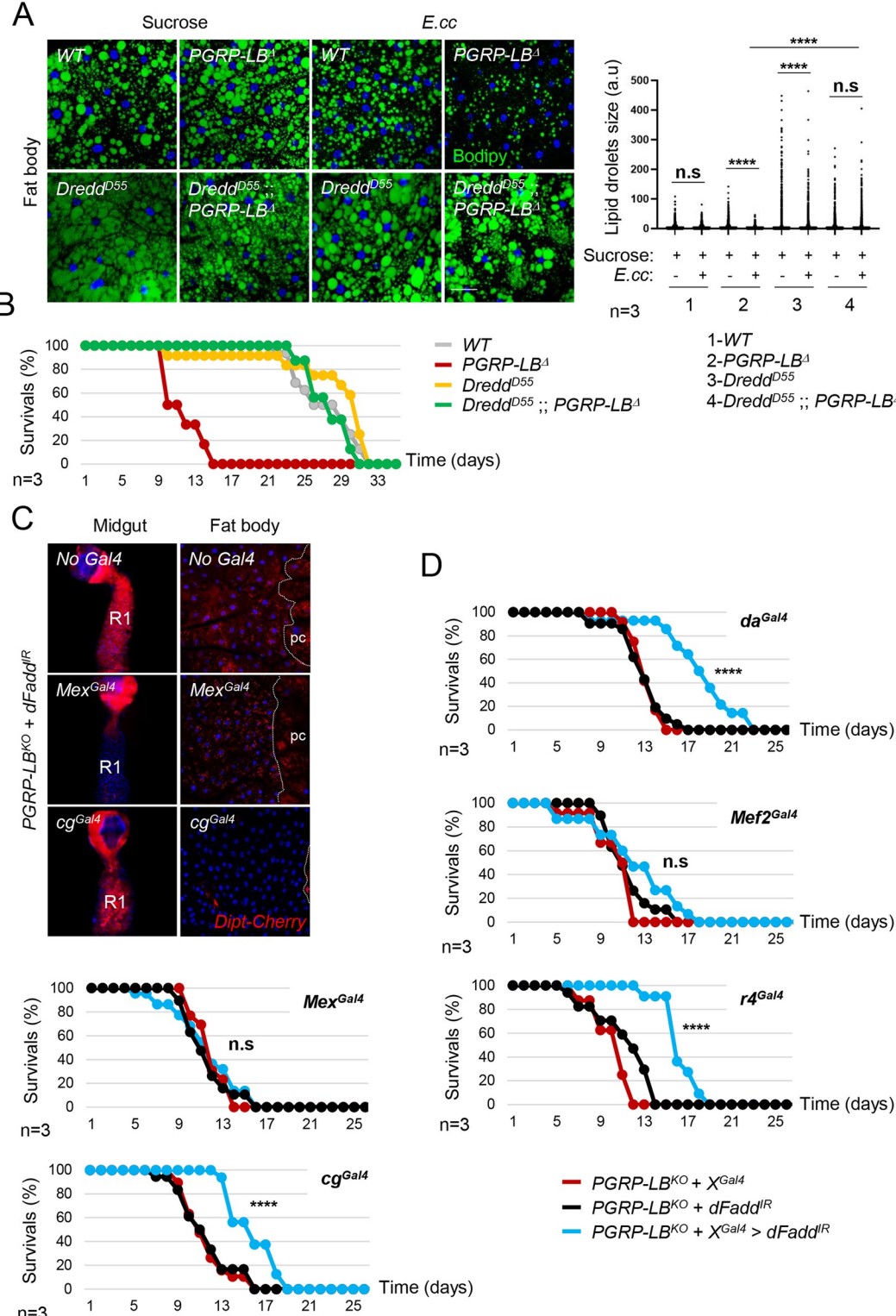

**Fig 6. Inhibition of the NF-κB signaling pathway in adipocytes ameliorates survival of *PGRP-LB^Δ* mutant flies infected with *E. cc*.** (A) (left panels) Confocal images of fat body from flies fed 8 days with sucrose or with a mixture of sucrose + *E. cc*, and showing lipid droplets labelled with Bodipy (green). The reduction of lipid storage characteristic of *E. cc* infected *PGRP-LB^Δ* flies is suppressed in presence of two copies of the loss of function allele *Dredd^D55*. (Right) Graph showing the quantification of lipid droplets size performed on confocal images obtained from 3 independent experiments (n = 3). a.u:

arbitrary unit. no significant (n.s), ****p<0.0001; Kruskal-Wallis test. (B) Survival analysis of flies orally infected with *E. cc*. The *Dredd$^{D55}$* allele suppresses the deleterious effect of *PGRP-LB$^\Delta$* mutation on fly's survival. For A and B, flies of the following genotypes were used: *w$^{1118}$*/*w$^{1118}$* (*Control*), *w$^{1118}$*/*w$^{1118}$*;; *PGRP-LB$^\Delta$*/*PGRP-LB$^\Delta$* (*PGRP-LB$^\Delta$*), *w$^{1118}$*, *Dredd$^{D55}$*/ *w$^{1118}$*, *Dredd$^{D55}$* (*Dredd$^{D55}$*) and *w$^{1118}$*, *Dredd$^{D55}$*/*w$^{1118}$*, *Dredd$^{D55}$*;; *PGRP-LB$^\Delta$*/*PGRP-LB$^\Delta$* (*Dredd$^{D55}$*;; *PGRP-LB$^\Delta$*). (C) Confocal images of the R1 domain of adult midgut (left panels) or of adult fat body (right panels) from *PGRP-LB$^\Delta$* mutant fed 24h with *E. cc*, showing *Dipt-Cherry* (red) expression. Inhibition of the IMD signaling pathway via expression of *UAS-dFadd$^{IR}$* is effective in enterocytes or in adipocytes using *Mex$^{Gal4}$* or *cg$^{Gal4}$*, respectively. (D) Survival analysis of *PGRP-LB$^\Delta$* mutant flies orally infected with *E. cc*. The expression of *UAS-dFadd$^{IR}$* ubiquitously, using *da$^{Gal4}$*, or in adipocytes, using either *cg$^{Gal4}$* or *r4$^{Gal4}$* decreases the deleterious effect of *PGRP-LB$^\Delta$* mutation on fly's survival. These flies have a significantly extended lifespan compared to the corresponding control flies (Log-rank (Mantel-Cox) test, ****p < 0.0001). Adult *PGRP-LB$^\Delta$* mutant flies expressing *UAS-dFadd$^{IR}$* in enterocytes, using either *Mex$^{Gal4}$* or in muscle, using *Mef2$^{Gal4}$* have no significant (n.s) extended lifespan compared to the corresponding control flies (Log-rank (Mantel-Cox) test). Flies of the following genotypes were used: *w$^{1118}$*/*w$^{1118}$*; +/+ or *Mex$^{Gal4}$*/+ or *cg$^{Gal4}$*/+; *PGRP-LB$^\Delta$*, *Dipt-Cherry$^{C1}$*/*PGRP-LB$^\Delta$*, *UAS-dFadd$^{IR}$* (C-D) and *w$^{1118}$*/*w$^{1118}$*;; *PGRP-LB$^\Delta$*, *da$^{Gal4}$* or *PGRP-LB$^\Delta$*, *Mef2$^{Gal4}$* or *PGRP-LB$^\Delta$*, *r4$^{Gal4}$*/*PGRP-LB$^\Delta$*, *UAS-dFadd$^{IR}$* (D). Scale bar is 100 μm (midguts in C) and 20 μm (fat bodies in A and C).

are unable to efficiently rescue adult's lifespan of undernourished flies [35]. Further investigation will be necessary to uncover the strain specific compounds and mechanism(s) responsible for this species-specific regulation of SREBP and, in a broader aspect, to stimulate lipid metabolism.

In *Drosophila*, immunity and metabolism are linked structurally via the fat body, an organ homologous to the mammalian liver, adipose tissue and immune system, made of a single cell type: the adipocyte [42]. In mammals, immune cells are embedded into the adipose tissue, allowing direct influence of one cell type on the other [7]. Our data indicate that chronic activation of the IMD/NF-κB pathway prevents gut bacteria-dependent SREBP processing and thus lipid metabolism. By restricting the diffusion of PGN to the fly hemolymph, the PGRP-LB$^{sec}$ enzyme allows gut bacteria-dependent lipogenesis in remote adipocytes and promote fly survival. In the absence of such brake, lipid storages of orally infected flies are rapidly depleted and life span is reduced. Since *E. cc*-fed *PGRP-LB$^\Delta$* mutant display ovarian atrophy associated with a reduction of vitellogenic stages [43], it is possible that of sex-hormones misregulation is contributing to SREBP processing inhibition in this organ.

We found that Foxo is a negative regulator of SREBP processing in adipocytes and that IMD/NF-κB signaling pathway inhibits SREBP processing, without affecting Insulin/PI3K signaling (S7 Fig). Thus, we propose that the NF-κB transcription factor Relish and Foxo acts in parallel, or together, to negatively regulates SREBP processing. Interestingly, both transcription factors have common immune and metabolic target genes in *Drosophila* fat body [6,10,44,45]. One possibility would be that Relish and Foxo negatively regulate the transcription of genes that are essentials for SREBP processing, such as the escort factor SCAP (SREBP Cleavage Activating Protein), and/or the proteases S1P (Site-1 Protease) and S2P (Site-2 Protease) [22,28].

Inhibition of lipid metabolism triggered by bacterial infection have been reported in the past, although in different contexts. When bacteria such as *Mycobacterium marinum* are injected into *Drosophila* body cavity, the transcription factor Mef2, which activates transcription of metabolic genes in non-infected individuals, switches its activity to enhance transcription of immune genes [8]. As a result, anabolic transcripts are reduced and energy stores, such as lipids, are lost. Toll and the IMD signaling pathways are acting genetically upstream of Mef2 in this process. Lee and colleagues found that *E. cc* infection triggers lipid catabolism in enterocytes which, via a TRAF3-AMPK/WTS-ATG1 pathway, contributes to the activation of DUOX, a member of the NADPH oxidase family acting as the first line of host defense in *Drosophila* gut [46]. Finally, the bacteria produced short chain fatty acid acetate acts as a microbial metabolic signal that activates signaling through the IMD pathway in enteroendocrine cells.

**Bacteria feeding**

**Fig 7. Model for the role of bacterial PGN and bacterial amino-acids in lipid storage formation.** In healthy flies, feeding bacteria under protein scarcity promotes bacterial amino-acids transfer from the gut into the hemolymph. This indirectly triggers Dilps secretion by neurosecretory cells of the central brain _leading to the systemic activation of the insulin/Tor signaling pathways and the activation of SREBP in adipocytes in order to sustain lipogenesis. Simultaneously, gut-born PGN diffuses into the hemolymph where it is degraded by the secreted amidase PGRP-LB[sec]. Upon chronic inflammation due to the lack of PGRP-LB[sec] and accumulation of PGN, the constitutive activation of the IMD signaling pathway promotes lipid depletion via inhibition of SREBP activation in adipocytes.

This, in turn, increases transcription of the endocrine peptide tachykinin, which is essential for timely larval development and optimal lipid metabolism and insulin signaling [47].

Our work sheds light on how gut bacteria influences lipid metabolism and contributes to the development of an immune-metabolic disorder, through the action of the highly conserved transcription factors SREBP, NF-κB and Foxo and the universal bacteria cell-wall component PGN. Furthermore, it shows how by buffering gut-born circulating PGN levels, the PGRP-LB amidase, allows the appropriate balance between metabolic and immune responses.

## Material and methods

### *Drosophila* strains and maintenance

The strains used in this work are: $w^{1118}$ BL#3605, *Gal4::SREBP* from BL#38395, *UAS-2XEGFP* BL#6874, *Diptericin-Cherry$^{C1}$* [48], *PGRP-LB$^{\Delta}$* [21], *PGRP-LC$^{E12}$* [49], *Rel$^{E20}$* BL#9457, *Dredd$^{D55}$* and *Dredd$^{F64}$* (a gift from François Leulier), *da$^{Gal4}$* BL#55851, *r4$^{Gal4}$* and *Mex$^{Gal4}$* (kindly provided by Yixian Zheng), *Mef2$^{Gal4}$* BL#27390, *cg$^{Gal4}$* BL#7011, *UAS-IMD* (kindly provided by François Leulier), *UAS-PGRP-LCa* BL#30917, *CoinFLP$^{Gal4}$* BL#59269, *hs-FLP.G5* BL#58356, *Tub$^{Gal80ts}$* BL#7108, *UAS-CD4::Tomato* (kindly provided by Frank Schnorrer), *LexA::SREBP* (this work, molecular details of the construct under request), *13XLexAop2-6XGFP* BL#52265, *LexAop-CD8-GFP-2A-CD8-GFP* BL#66545, *tGPH* BL#8163, *13XLex-Aop2-mcd8-GFP* BL#32203, *chico$^{1}$* BL#10738, *Df(2L)ED729* BL#24134, *foxo$^{\Delta94}$* BL#42220, *foxo$^{25}$* BL#80944, *PGRP-LB$^{PD*Z10e}$* and *PGRP-LB$^{PC*10A}$* (Kurz et al., 2017), *UAS-dFadd$^{IR}$* ([50]; kindly provided by Pascal Meier), *UAS-InR$^{DN}$* BL#8252 and *UAS-ATG1$^{IR}$* BL#44034. Flies were grown at 25˚C on a yeast/cornmeal medium in 12h/12h light/dark cycle-controlled incubators. For 1liter of food, 8.2g of agar (VWR, cat. #20768.361), 80g of cornmeal flour (Westhove, Farigel maize H1) and 80g of yeast extract (VWR, cat. #24979.413) were cooked for 10 min in boiling water. 5.2 g of Methylparaben sodium salt (MERCK, cat. #106756) and 4 ml of 99% propionic acid (CARLOERBA, cat. #409553) were added when the food had cooled down. For antibiotic (ATB) treatment, the standard medium was supplemented with Ampicillin, Kanamycin, Tetracyclin and, Erythromycin at 50 μg/ml final concentrations.

### *Drosophila* genetics and analysis

To generate *UAS-IMD* and *UAS-PGRP-LCa* overexpressing clones or to overexpress *UAS-InR$^{DN}$* in adult adipocytes or *UAS-ATG1$^{IR}$* in adult enterocytes, 5 days old mated females were raised and aged in presence of ATB at 22˚C. Adult flies were then transferred into non-ATB media, and placed 24h at 29˚C to inactivate Gal$^{80ts}$, before the infection by bacteria for the following 48h at 29˚C. For *UAS-IMD* and *UAS-PGRP-LCa* overexpressing clones, no heat shock was required for clone induction. Flies of the following genotype were used: *Coin-FLP$^{Gal4}$/+; hs-FLP.G5/LexA::SREBP, LexAop-CD8-GFP-2A-CD8-GFP; UAS-IMD, Tub$^{Gal80ts}$/+ or PGRP-LCa, Tub$^{Gal80ts}$/+* for Fig 5D or *CoinFLP$^{Gal4}$/+; hs-FLP.G5/tGPH, UAS-CD4::Tomato; UAS-IMD, Tub$^{Gal80ts}$/+ or PGRP-LCa, Tub$^{Gal80ts}$/+* for S4B Fig.

### Imaging

Whole fly imaging was performed on adult females totally immersed in 70% EtOH. Images were captured using a ZEISS SteREO Discovery.V12 microscope. For dissected tissues, adult flies were cut apart in cold PBS, fixed for 20 min in 4% paraformaldehyde on ice and rinsed 3 times in PBT (1XPBS + 0.1% Triton X-100). The tissues were mounted in Vectashield (Vector Laboratories) fluorescent mounting medium, with or without DAPI. Images were captured with an LSM 780 ZEISS confocal microscope.

### Bodipy and Nile red staining

For Bodipy and Nile Red staining, adult tissues were dissected in PBS, fixed for 20 min in 4% paraformaldehyde on ice, rinse 3 times in 1XPBS without detergent and stained with Nile red (Cat. No. 72485, Sigma-Aldrich) or Bodipy 493/503 (Cat. No. D3922, ThermoFisher) at respectively 1:10000 and 1:1000, in PBS for 30 min.

### Immunofluorescence

Adult flies were cut apart in cold PBS, fixed for 20 min in 4% paraformaldehyde on ice and rinsed 3 times in PBT (1XPBS + 0.1% Triton X-100). Dissected tissues were blocked 2h in PBT + 3% BSA and then incubated overnight with the primary anti-dSREBP (3B2, Ref. ATCC-CRL-2693 from ACC) antibody (1:50 in PBT), in a cold room, without shaking. After 3 washes in PBT, the dissected tissues were incubated 2h with a goat anti-mouse Alexa Fluor 488 antibody (1:500 in PBT, Ref. ab150113 from abcam). The tissues were next rinsed three times in PBT and mounted in Vectashield (Vector Laboratories) fluorescent mounting medium, with DAPI. Images were captured with an LSM 780 ZEISS confocal microscope.

### Quantification of lipid droplets

Fiji/ImageJ was used for quantification of lipid droplet size in adult adipocytes imaged by confocal microscopy. First, the area was measured for the entire field of view or a region-of-interest (ROI) of defined size. Second, and prior to converting image to binary, a 'smooth' function was applied to the image to remove inherent graininess of the lipid stain and allow for more accurate quantification of lipid droplets. Then 'Watershed' was performed on binary images, and 'analyze particles' was used to quantify LD number and size.

### Bacterial strains

The following microorganisms were used: *Erwinia carotovora carotovora 15* strain 2141 (grown at 30˚C), *Lactobacillus plantarum* strain WJL (grown at 37˚C), *Escherichia coli* strain DH5$\alpha$ (grown at 37˚C), *Acetobacter pomorum* (grown at 30˚C) and *Enterococcus faecalis* (grown at 37˚C). Microorganisms were cultured overnight in Luria-Bertani (for *E. cc*, *E. coli* and *E. faecalis*) or MRS medium (for *L.plantarum* and *A. pomorum*). Cultures were centrifuged at 4000 g for 15 min at RT and re-suspended in 1XPBS. Cells were serially diluted in 1XPBS and their concentration was determined by optical density (OD) measurement at 600 nm. For heat killed bacteria, cells were re-suspended in 1XPBS and heated at 95˚C for 15 min.

### Adult oral infection

We used 4–6 days old female raised at 25˚C in presence of ATB in the food. 24h before the infection, female flies were transferred in vials without ATB and then placed in a fly vial with microorganism contaminated food. The food solution was obtained by mixing a pellet of an overnight culture of bacteria or yeast (OD = 200) with a solution of 5% sucrose (50/50) and added to a filter disk that completely covered the agar surface of the fly vial. For *E. coli* heat inactivation, a solution of *E. coli* diluted (final $OD_{600}$ = 100) in 2,5% Sucrose was incubated at 96˚C for 20 minutes, then cool down before use.

### Survival tests with bacterial infection

For oral infections, 15 adult flies were transferred every 2 days in a fresh vial in which 150 microliters of a fresh solution of *E. cc* (OD = 200)/ 5% sucrose (50/50) has been deposited. Experiments were performed in triplicate.

### Quantitative real-time PCR

RNA from whole dissected organs (n = 12) was extracted with RNeasy Mini Kit. Three hundred ng of total-RNA was then reverse transcribed in 10 μl reaction volume using the Superscript III enzyme (Invitrogen) and random hexamer primers. Quantitative real-time PCR was performed on a CFX96 Real-Time PCR Detection System (BIO-RAD) in 96-well plates using

the FastStart Universal SYBR Green Master (Sigma-Aldrich). The amount of mRNA detected was normalized to control rp49 mRNA values. Normalized data was used to quantify the relative levels of a given mRNA according to cycling threshold analysis ($\Delta$Ct). All datasets were organized and analyzed in Microsoft Excel 2016.

## Plasmid *pP{LexA:: dSREBPg}*

An 8.7 kb fragment (containing the entire dSREBP gene, 2.9 kb upstream and 0.7 kb downstream) was amplified by PCR using the High-Fidelity PCR System (Roche) and the P[pacman] BAC CH322-183B11 as DNA template. The forward primer used for amplification was 5'-CGGAATTCCGCATGCTCCCAGAGATGGCACTTTGG-3' and the reverse primer was 5'-GCGAATTCCACATGTCATCACTGTCAGCGGGATACC-3'. EcoR1 linkers were added during amplification and the resultant fragment was ligated into pRIV^white (a gift from Jean-Paul Vincent) to obtain pP{dSREBPg}. The open reading frame was sequenced in its entirety. Restriction sites for Asc1 and Fse1 were inserted into pP{dSREBPg} at the beginning of the ORF (Asc1, inserted immediately after aa3) and immediately following the bHLH region (Fse1, inserted immediately preceding aa. 362). The primers used for insertion of the Asc1 site were 5'-GCAGCATTCGCAATGGACACGGCGCGCCTGAACTTAATAGACGCT-3' and its reverse complement. Primers used for insertion of the Fse1 site were 5'-GCGACGGCTCC AAGGTGAAGGCCGGCCTTCAGCTGGGCACTCGGC-3' and its reverse complement. The sites were inserted individually into pP{dSREBPg}. Nar1 (for the Asc1 site) or Nar1-Nhe1 fragments (for the Fse1 site) were excised out of the resultant vector and then subcloned together into Nar1-Nhe1 digested pP{dSREBPg}. The resultant vector pP{dSREBPg/AF} was sequenced in the regions that had been subject to PCR. In order to generate pP{LexA::SREBP}, a cDNA fragment encoding a fusion of the LexA DNA binding domain fused to the RelA transactivation domain was amplified by PCR from pBPnlsLexA::p65Uw (Plasmid #26230 from Addgene). Asc1 and Fse1 linkers were added during amplification. This fragment was then ligated into pP{dSREBPg/AF}.

## Statistical analysis

The Prism software (GraphPad) was used for statistical analyses. For q-RT PCR experiments we used the nonparametric Kruskal-Wallis test. P value was indicated as follow: * for P<0,05, ** for P<0,01, *** for P<0,001. ns for not significantly different. We used the log-rank test Mantel-Cox for survival data analyses. **** for P<0,0001. ns for not significantly different.

## Supporting information

**S1 Fig. *Gal4::SREBP* and *LexA::SREBP* transgenes used in this study.** 1, Schematic drawing of the *SREBP* genomic locus. 2, In the *Gal4*::*SREBP* transgene, the transcription factor domain-encoding sequence was replaced by a Gal4::VP16-encoding sequence to report SREBP activation. 3, In the *LexA*::*SREBP* transgene the transcription factor domain-encoding sequence was replaced by a LexA::RelA-encoding sequence to report SREBP activation. (TIF)

**S2 Fig. Gut bacteria stimulate *LexA::SREBP* activation in adult fat body.** (A) Pictures of adult flies fed 2 days with sucrose, or a mixture of sucrose + *E. coli* or *E. cc*, showing *LexA*:: *SREBP* activation (green). Flies fed with sucrose show activation of *LexA*::*SREBP* in oenocytes, noticeable after a longer exposure time (panel with asterisk). Both *E. coli* and *E. cc* feeding promotes activation of *LexA*::*SREBP* in fat bodies. (B) Confocal images of fat body from flies fed 2 days with sucrose or with *E. coli* and showing *LexA*::*SREBP* activation (green). Flies fed on

sucrose show feeble activation of *LexA*::*SREBP* in adipocytes, noticeable after increasing the gain during image acquisition (panel with asterisk). *E. coli* feeding, however, promotes strong activation of *LexA*::*SREBP* in adipocytes. (C) Pictures of adult flies, control or *PGRP-LB^Δ^* mutant, fed 2 days with sucrose + *E. cc*, showing *LexA*::*SREBP* activation (green). Ingestion of *E. cc* triggers activation of *LexA*::*SREBP* in fat body from control flies, but not from *PGRP-LB^Δ^* mutant's flies. Flies of the following genotypes were used: *w^1118^/w^1118^*, *LexA*::*SREBP*, *13XLexAop2-6XGFP/+* (*Control* in A, B and C), and *w^1118^/w^1118^*, *LexA*::*SREBP*, *13XLexAop2-6XGFP/+; PGRP-LB^Δ^/PGRP-LB^Δ^* (*PGRP-LB^Δ^* in C). Scale bar is 0,25 mm (A and C) and 20 μm (B).
(TIF)

**S3 Fig. Gut bacteria sustain oogenesis.** Quantification of the different stages of oocytes observed in female's ovary, after feeding 24h on sucrose, or on a mixture of sucrose + *E. coli* or *E. cc*. Apoptotic events were quantified as oocytes with compact and dense nurse cell nuclei, using DAPI staining. Histograms correspond to the mean value ± SD of three experiments (n = 3). For each oocyte stage, sucrose values were used as reference for statistical analysis. *p<0.05; Kruskal-Wallis test. Flies of the following genotypes were used: *w^1118^/w^1118^*; *LexA*::*SREBP*, *13XLexAop2-6XGFP/+*.
(TIF)

**S4 Fig. PGRP-LC and Relish are required for SREBP inhibition observed in *E. cc* orally infected *PGRP-LB* mutants.** Confocal images of fat body from flies fed 2 days with a mixture of sucrose + *E. cc*. Double mutant *PGRP-LC^E12^*, *PGRP-LB^Δ^* or *Rel^E20^*, *PGRP-LB^Δ^* flies activate *Gal4*::*SREBP* in adipocytes from flies fed 2 days with *E. cc*, while *PGRP-LB^Δ^* mutants do not. Flies of the following genotypes were used: *w^1118^/w^1118^*; *Gal4*::*SREBP*, *UAS-2XEGFP/+; PGRP-LC^E12^*, *PGRP-LB^Δ^ / PGRP-LB^Δ^* (Top left panel, *PGRP-LB^Δ^*) or *w^1118^/w^1118^*; *Gal4*::*SREBP*, *UAS-2XEGFP/+; PGRP-LC^E12^*, *PGRP-LB^Δ^ / PGRP-LC^E12^*, *PGRP-LB^Δ^* (*PGRP-LC^E12^*, *PGRP-LB^Δ^*) or *w^1118^/w^1118^*; *Gal4*::*SREBP*, *UAS-2XEGFP/+; Rel^E20^*, *PGRP-LB^Δ^ / PGRP-LB^Δ^* (Top right panel, *PGRP-LB^Δ^*) or *w^1118^/w^1118^*; *Gal4*::*SREBP*, *UAS-2XEGFP/+; Rel^E20^*, *PGRP-LB^Δ^ / Rel^E20^*, *PGRP-LB^Δ^* (*PGRP-LC^E12^*, *PGRP-LB^Δ^*). Scale bar is 20 μm.
(TIF)

**S5 Fig. Activation of SREBP by heat killed bacteria or by *E. coli* on regular food.** (A) Confocal images of fat body from female flies fed 2 days with a mixture of sucrose + alive or heat killed bacteria (*E. coli* or *E. cc*), showing *LexA*::*SREBP* activation (green). Both heat killed bacteria are efficiently activating SREBP in adipocytes. (B) Confocal images of fat body from female flies fed 2 days with *E. coli* without sucrose, or with *E. coli* dropped on regular food, showing *Gal4*::*SREBP* activation (green). Absence of sucrose does not impact the strong activation of SREBP by *E. coli*, while presence of regular food diminishes it. Flies of the following genotypes were used: *w^1118^/w^1118^*; *LexA*::*SREBP*, *13XLexAop2-6XGFP/+* (A) and *w^1118^/w^1118^*; *Gal4*::*SREBP*, *UAS-2XEGFP/+* (B). Scale bar is 20 μm.
(TIF)

**S6 Fig. Survival curves of PGRP-LB mutans fed with sucrose.** Survival analysis of *PGRP-LB^Δ^* mutant flies fed with sucrose. The expression of *UAS-dFadd^IR^* ubiquitously using *da^Gal4^*, or in adipocytes, using either *cg^Gal4^* or *r4^Gal4^*, or in enterocytes using *Mex^Gal4^* or in muscle, using *Mef2^Gal4^* have no significant impact on flies' lifespan, compared to the corresponding control flies. Flies of the following genotypes were used: *w^1118^/w^1118^*;; *PGRP-LB^Δ^*, *da^Gal4^* or *PGRP-LB^Δ^*, *Mef2^Gal4^* or *PGRP-LB^Δ^*, *r4^Gal4^/PGRP-LB^Δ^*, *UAS-dFadd^IR^*.
(TIFF)

**S7 Fig. Gut bacteria promote cell surface recruitment of tGPH in adipocytes and this is not affected by over expressing IMD or PGRP-LCa cell autonomously.** Confocal images of fat body from adult flies fed 2 days with either sucrose, or a mixture of sucrose + *E. faecalis* or heat killed (H.k) *E. coli*, and showing the *tGPH* marker (green). Ingestion of heat killed *E. coli* promote recruitment of *tGPH* at the cell surface of adipocytes, compared to a sucrose diet. (B) Confocal images of fat body showing clones of adipocytes overexpressing IMD or PGRP-LCa (red) and the *tGPH* marker (green), from flies fed 1 day with *E. cc* or 2 days with *E. coli*. Fat body clones over expressing either IMD or PGRP-LCa do not affect *tGPH* recruitment at the cell surface of adipocytes. Flies of the following genotypes were used: $w^{1118}/w^{1118}$; *tGPH/tGPH* (A) and $w^{1118}$, *CoinFLP$^{Gal4}$/$w^{1118}$*; *tGPH, UAS-CD4::Tomato/hs-FLP.G5, Tub$^{Gal80ts}$*; *UAS-IMD or UAS-PGRP-LCa/+*. Scale bar is 20 μm.
(TIFF)

**S1 Numerical data. Numerical data underlying the results.**
(XLSX)

## Acknowledgments

We thank Sabine Peslier for technical help.

## Author Contributions

**Conceptualization:** Bernard Charroux, Julien Royet.

**Formal analysis:** Bernard Charroux.

**Funding acquisition:** Julien Royet.

**Investigation:** Bernard Charroux.

**Visualization:** Bernard Charroux.

**Writing – original draft:** Bernard Charroux.

**Writing – review & editing:** Bernard Charroux, Julien Royet.

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
