## [Decision Letter · Decision Letter 0]

26 Aug 2021

Dear Dr Royet,

Thank you very much for submitting your Research Article entitled 'Gut-born peptidoglycan remotely inhibits gut-bacteria dependent activation of SREBP by Drosophila adipocytes' to PLOS Genetics.

The manuscript was fully evaluated at the editorial level and by independent peer reviewers. The reviewers appreciated the attention to an important problem, but raised some substantial concerns about the current manuscript. Based on the reviews, we will not be able to accept this version of the manuscript, but we would be willing to review a much-revised version. We cannot, of course, promise publication at that time.

If you decide to revise the manuscript for further consideration at PLOS Genetics, please aim to resubmit within the next 60 days, unless it will take extra time to address the concerns of the reviewers, in which case we would appreciate an expected resubmission date by email to plosgenetics@plos.org.

[LINK]

We are sorry that we cannot be more positive about your manuscript at this stage. Please do not hesitate to contact us if you have any concerns or questions.

Yours sincerely,

Gregory P. Copenhaver, Ph.D.

Editor-in-Chief

PLOS Genetics

Reviewer's Responses to Questions

**Comments to the Authors:**

Reviewer #1: In this study, Charroux and Royet investigated the molecular mechanism by which gut bacteria modulate lipid momeostasis via cross-regulation between Insulin-SREBP signaling and PGN-IMD signaling. They first observed that specific gut bacterial species including E. coli and Erwinia carotovora carotovora (ECC) could activate an SREBP-dependent lipogenesis in enterocyte, fat body and ovaries. They next showed that insulin singaling activation is requried for bacterial-induced SREBP activation. Interestingly, bacterial-induced SREBP activation is abolished under PGRP-LB mutant backgroud, which could be reverted by the simultanous IMD pathway inactivation, demonstrating that IMD pathway activation is able to antagonize bacterial-induced SREBP activation. Clonal analysis further confirmed that bacterial-induced SREBP activation in fat body cells is repressed by forced activation of IMD pathway in a cell –autonomous manner. They observed that PGRP-LB mutant animals showed reduced lifespan upon bacterial infection. Importantly, tissue-specific inactivation of IMD pathway in bacterial-infected PGRP-LB mutant animals revealed that inactivation of IMD pathway in fat body, but not muscle or enterocytes, is sufficient to improve the surivival of PGRP-LB mutant animals. Taken together, they concluded that cross-regulation between IMD-NF-kappaB pathway and Insulin-FOXO and SREBP pathway is required for lipid homeostasis and host survival during enteric infection, highlighting the importance of a balance between metabolic and immune signaling.

General comments.

Immune-metabolic cross-talk is an important issue, and plays a pivotal role in physiological adaptation at an organismal level during inflammation. However, at present, the precise mechanism gourvenring this cross-talk is poorly understood. In this context, I found that this study is novel and important in the field. They used an elegant genetic approach to clearly demonstrate their hypothesis and many of the experiments are meticulously controlled. I have only few comments and suggestions that would improve the significances of the present work.

Major Point

Perhaps, the metabolic changes during the course of infection “might” be dynamic. For example, in the early phase of infection (e.g. within 30 min to few hr), the insulin signaling may be somewhat inactivated or down-regulated (possibly by infection-induced NF-kappaB pathway activation) and lipogenesis is blocked. This process may be required for allocating energy for the immune response and/or for the inhibition of excess ROS production. For example, a recent paper (Immunity, 2020, 52, 374pp) showed that septic infection rapidly induces the excretion of hemolymph lipid to prevent ROS production. They showed that lipid level started to be depleted 1 hr after infection, and remained at a very low level over 6 hr (less than 10% compared with the level of control animal!), and then started to be replenished (that “may” certainly require lipogenesis in adipocytes) and reached to normal level at 24hr. They found a similar decrease of lipids after oral bacterial infection with ECC (see figure S1B of Immunity, 2020). Therefore, it may be possible that animal uses or excretes lipids while preventing lipogenesis, at least in an early time point of infection. However, in the late phase of infection (~12 hr or later), animals may actively operate insulin signaling to promote lopogenesis for the lipid homeostasis (e.g. in part to replenish the hemolymph lipid). In this context of senario, it is interesting to discuss these important points in the discussion section or to examine time-course activation of insulin signaling because the authors examined insulin activation (SREBP activation) mainly in the late phase of infection (~ few days following infection).

Minor points

1. Why is “adipocyte-specific” lipogenesis (SREBP activation) important for host survival following gut infection? It might be interesting to discuss this issue in the disucssion section.

2. Is forced insulin signaling activation (e.g. InR overexpression or FOXO inactivation) in the fat body of PGRP-LB mutants sufficient to confer host resistance to bacterial infection?

Reviewer #2: In this ms, Charroux and Royet describe a systemic circuitry that allows metabolic adaptation in response to oral bacterial infection in Drosophila. They propose that, upon infection, SREBP activation and lipid droplets are induced in adipocytes, by coordinating insulin signaling induction with blocking, via PGRP-LB-dependent inactivation of bacterial PGN, of IMD-dependent SREBP inhibition. PGRP-LB loss antagonizes insulin-dependent SREBP activation, leading to chronic inflammation and premature fly death. Overall, the findings provide interesting insights in adipocytes immune-metabolic regulation, are clearly presented and experiments are well conducted. However I do have some concerns that should be addressed to support the conclusions.

Major comments:

- the authors’ model is consistent with a physiological response to infection that favors anabolism vs catabolism in adipocytes. How this response is induced should be experimentally clarified. For example, if the authors are proposing that this response is triggered, as suggested by their model, to balance protein catabolism induced in the gut by bacterial infection, this idea should be experimentally tested in their settings. One possibility would be that blocking bacteria-induced protein catabolism in the gut prevents SREBP and lipid droplets induction in adipocytes. Furthermore, the implied requirement of DILPs for insulin signaling activation in adipocytes should be experimentally assessed in the authors’ settings. It would also be important to more generally assess whether catabolic pathways are inhibited in adipocytes of infected flies, for example those dependent on the activity of AMPK, which indeed was suggested to inhibit SREBP in Drosophila adipocytes (Bertolio et al. Nat Comm 2019).

- in bacterial-infected PGRP-LB mutants, the requirement of PGRP-LC and other IMD pathway components (tak1, rel, ikk) for inhibition of SREBP activation, beside Dredd and dFadd, should be assessed. It is unclear why different dredd alleles where used in figs5/6. Both should be tested in each experiment. Related to this, rescue of fly survival by dFadd RNAi in fig6D is mild compared to fig6B. Is this due to pathway redundancy? What is the efficiency of dFadd RNAi ?

- in most experiments, SREBP activation is assessed just with the LexA::SREBP reporter activation. Why the pattern of LexA::SREBP reporter activation (fig2a,b) in ovaries is different from that of Gal4::SREBP reported Kunte et al (ref 22)? LexA::SREBP reporter activation in key experiments should be paralleled by immunofluorescence to detect nuclear localization of endogenous SREBP and induction of SREBP targets. Relate to this, the effect of PGRP-LB mutation reducing SREBP activation, as quantified with ACS induction in fig 1C, is unclear. Is the difference in ACS induction between control and PGRP-LB mutant statistically significant? Also, lipid droplets in fat bodies should be quantified.

Minor issues:

- all figure panels should include controls, whose lack often obliges the reader to pick them in other figures (just as examples, fig5a, NF-kB activation with PGRP-LCa or IMD overexpression, lifespan of non-infected flies in fig6B);

- all figure panels should be referenced to in the text (for example fig.S4d).

**Have all data underlying the figures and results presented in the manuscript been provided?**

Reviewer #1: Yes

Reviewer #2: Yes

PLOS authors have the option to publish the peer review history of their article (what does this mean?). If published, this will include your full peer review and any attached files.

Reviewer #1: No

Reviewer #2: No

---

## [Decision Letter · Decision Letter 1]

14 Feb 2022

Dear Dr Royet,

We are pleased to inform you that your manuscript entitled "Gut-derived peptidoglycan remotely inhibits bacteria dependent activation of SREBP by Drosophila adipocytes" has been editorially accepted for publication in PLOS Genetics. Congratulations!

Yours sincerely,

Gregory P. Copenhaver

Editor-in-Chief

PLOS Genetics

Comments from the reviewers (if applicable):

Reviewer's Responses to Questions

**Comments to the Authors:**

Reviewer #1: I am satisfied with the revision, and I feel that the manuscript is adequate for the publication.

Reviewer #2: The authors have now addressed most of my concerns and i think that new data reinforce the conclusions. I do not have any more comments.

**Have all data underlying the figures and results presented in the manuscript been provided?**

Reviewer #1: Yes

Reviewer #2: Yes

PLOS authors have the option to publish the peer review history of their article (what does this mean?). If published, this will include your full peer review and any attached files.

Reviewer #1: No

Reviewer #2: No

**Data Deposition**

http://datadryad.org/submit?journalID=pgenetics&manu=PGENETICS-D-21-00871R1

**Press Queries**

---

## [Editor Report · Acceptance letter]

1 Mar 2022

PGENETICS-D-21-00871R1 

Gut-derived peptidoglycan remotely inhibits bacteria dependent activation of SREBP by Drosophila adipocytes 

Dear Dr Royet, 

We are pleased to inform you that your manuscript entitled "Gut-derived peptidoglycan remotely inhibits bacteria dependent activation of SREBP by Drosophila adipocytes" has been formally accepted for publication in PLOS Genetics! Your manuscript is now with our production department and you will be notified of the publication date in due course.

With kind regards,

Agnes Pap

PLOS Genetics

On behalf of:
